# Treatment Strategy for Multiple Myeloma to Improve Immunological Environment and Maintain MRD Negativity

**DOI:** 10.3390/cancers13194867

**Published:** 2021-09-28

**Authors:** Kazuhito Suzuki, Kaichi Nishiwaki, Shingo Yano

**Affiliations:** 1Department of Internal Medicine, Division of Clinical Oncology and Hematology, The Jikei University Kashiwa Hospital, Tokyo 277-8567, Japan; nishiwaki@jikei.ac.jp; 2Department of Internal Medicine, Division of Clinical Oncology and Hematology, The Jikei University School of Medicine, Tokyo 105-8461, Japan; yano@jikei.ac.jp

**Keywords:** multiple myeloma, immune environment, minimal residual disease, proteasome inhibitor, immunomodulatory drug, monoclonal antibody, autologous stem cell transplantation

## Abstract

**Simple Summary:**

Improving the immunological environment and eradicating minimal residual disease (MRD) are the two main treatment goals for long-term survival in patients with multiple myeloma (MM). An improved immunological environment may be useful for maintaining MRD negativity. Whether the ongoing treatment should be continued or changed if the MRD status remains positive is controversial. In this case, genetic, immunophenotypic, and clinical analysis of residual myeloma cells may be necessary to select the effective treatment for the residual myeloma cells. The purpose of this review is to discuss the MM treatment strategy to “cure MM” based on currently available therapies and expected immunotherapies via improvement of the immunological environment and maintenance of MRD negativity.

**Abstract:**

Improving the immunological environment and eradicating minimal residual disease (MRD) are the two main treatment goals for long-term survival in patients with multiple myeloma (MM). Immunomodulatory drugs (IMiDs), monoclonal antibody drugs (MoAbs), and autologous grafts for autologous stem cell transplantation (ASCT) can improve the immunological microenvironment. ASCT, MoAbs, and proteasome inhibitors (PIs) may be important for the achievement of MRD negativity. An improved immunological environment may be useful for maintaining MRD negativity, although the specific treatment for persistent MRD negativity is unknown. However, whether the ongoing treatment should be continued or changed if the MRD status remains positive is controversial. In this case, genetic, immunophenotypic, and clinical analysis of residual myeloma cells may be necessary to select the effective treatment for the residual myeloma cells. The purpose of this review is to discuss the MM treatment strategy to “cure MM” based on currently available therapies, including IMiDs, PIs, MoAbs, and ASCT, and expected immunotherapies, such as chimeric antigen receptor T cell (CAR-T) therapy, via improvement of the immunological environment and maintenance of MRD negativity.

## 1. Introduction

Multiple myeloma (MM) is a hematopoietic malignancy of the plasma cells, and although the survival of patients with MM has been prolonged by the development of new agents in the last few decades, it is still an incurable disease [1,2].

To cure MM, it is important to improve the immune environment and ensure persistent minimal residual disease (MRD) negativity [3,4,5]. Notably, the immune environment of myeloma patients is characterized by an attenuated immune effect on tumor cells, creating an environment suitable for the survival of myeloma cells [3,4]. However, an improved immune environment leads to the long-term survival of patients with myeloma due to enhanced immunological potency against myeloma cells [6]. Recently, various immunotherapeutic agents, including immunomodulatory drugs (IMiDs) and monoclonal antibody drugs (MoAbs) against CD38 and signaling lymphocytic activation molecule family 7 (SLAMF7), have been developed [7,8,9,10,11]. In addition, the clinical development of an immune checkpoint inhibitor for myeloma, which has played an important role in the treatment of solid malignant tumors, is under way [12]. Autologous grafts used in autologous stem cell transplantation (ASCT), which is still the standard treatment for patients with MM [13,14], have been reported to improve the immune environment [15].

MRD-negativity, which is analyzed using next-generation sequencing (NGS) and next-generation flow cytometry (NGF), prolongs the progression-free survival (PFS) and overall survival (OS) of patients [5]. Persistent MRD negativity in multiple assessments is important for long-term survival [5]. However, the prognosis of MRD-positive patients is not good, even if complete response (CR) is achieved. Therefore, eradicating all myeloma cells should be the primary treatment goal for MRD-positive patients, although sustained MRD-positivity is not always an unfavorable outcome [16]. Genetic and immunophenotypic characterization of residual myeloma cells, including the clinical course, can be essential for defining and selecting a suitable treatment strategy.

The purpose of this review is to describe the importance of improving the immune environment in MM patients and its therapeutic strategies, the clinical significance of MRD status for long-term survival, and therapeutic strategies for persistent MRD negativity. We also describe the treatment of residual myeloma cells in MRD-positive patients and the future MRD status-adapted treatment strategies.

## 2. Immunological Environment in MM

The immune system plays an important role in the genesis of myeloma. The functions of immune cells are suppressed by cytokines and the interaction between myeloma cells and the bone marrow (BM) microenvironment [17,18]. A potential positive relationship between the cellular components of the immune system, such as T cells, natural killer (NK) cells, regulatory T cells (Treg), and B cells, and myeloma progression was suggested in previous studies [17,18,19]. According to an earlier report, disease status, advanced stage in the International Staging System (ISS), and high-risk cytogenetic abnormalities (HRCA) were related to worse immune profiles [18].

T cells are categorized into cytotoxic CD8^+^ T cells and helper CD4^+^ T cells. Cytotoxic T cells (CTL) are effector cells for adoptive immune responses and are activated by interleukin (IL)-2 and exert their anti-tumor effect by releasing interferon-gamma following antigen presentation [19]. CD4^+^ T cells mainly enhance the adaptive immune response [20]. T-cells are quantitatively and functionally altered in MM and, consequently, have a role in the immunodeficiency associated with myeloma pathogenesis [21]. The frequencies of effector memory and effector CD8^+^ T cells in MM patients are higher than those in healthy individuals, while the frequency of CD4^+^ T cells is similar between MM patients and healthy individuals [22]. Low CD4^+^ T cell counts and low CD4/CD8 ratios in peripheral blood (PB) are predictors of poor clinical outcomes [23].

T cells, especially in tumor sites, are exhausted in patients with MM compared to those with monoclonal gammopathy of undetermined significance (MGUS) [22,24,25]. T cell exhaustion is induced by inadequate IFN-gamma and upregulation of inhibitory receptors on T cells, such as programmed cell death protein1 (PD-1), cytotoxic T-lymphocyte-associated antigen 4 (CTLA-4), Tim-3, and lymphocyte activation gene-3 (LAG-3), in MM patients [24,26]. Galectin-9 and a proliferation-inducing ligand (APRIL) derived from osteoclasts, which constitute the microenvironment of myeloma cells, induce Tim-3 and programmed death-ligand 1 (PD-L1) on MM cells, respectively, and contribute to immune escape [27]. Soluble PD-L1, derived from MM cells, suppresses the immune system by binding to PD-1 on CTL [28]. T cell anergy is a tolerance mechanism due to the inactivation of lymphocytes, and anergic T cells remain alive for an extended period in a hyporesponsive state [29]. Anergic T cells are induced by co-stimulation of the T cell receptor (TCR) and low expression of CD28 and high expression of CTLA-4 on MM cells [24,30]. The positivity of PD-1 and CTLA-4 on both CD4^+^ and CD8^+^ T cells in MM patients was higher than that in healthy individuals in BM [24]. PD-1^+^ lymphocytes contribute to the proliferation of functionally impaired tumor-specific lymphocytes [31]. The high frequency of PD-1- or CTLA-4-expressing CD8^+^ T cells was not significantly different in MM patients before and after IMiDs treatment [22]. In addition, PD-1^+^CD38^+^ lymphocytes suppress anti-cancer activity and have been identified in patients with malignancies [32], especially after administration of anti-PD-1 MoAbs [33]. Thus, PD-1 and PD-L1 are therapeutic targets in MM.

NK cells are effector lymphocytes for the innate immune response, control several types of tumors and infections, and regulate the activities of T cells, macrophages, and dendritic cells [34]. Elevated NK cell counts in PB and BM are noted in patients with early-phase myeloma, but the number of NK cells in PB decreases as myeloma progresses. Moreover, NK cell activity is reduced in patients with MM [35]. In antibody-dependent cellular cytotoxicity (ADCC) activity induced by the binding of Fcγ receptors to the Fc tail of the MoAbs, NK cells release toxic proteins, including granzymes and perforins, which kill myeloma cells [36]. However, myeloma cells have reduced ADCC activity due to downregulated expression of NK cell receptors, such as natural killer group 2D (NKG2D), NKp30, CD244, and DNAX accessory molecule 1 (DNAM-1) [37,38]. In addition, the expression of PD-1 on NK cells prevents immune recognition of tumor cells in myeloma patients [39]. ADCC activity can be induced by several MoAbs, including PD-1 blockade, and is essential for MM treatment.

Treg cells comprise 5–7% of CD4^+^ T cells and develop from CD4^+^ T cells under conditions of high levels of transforming growth factor-β (TGF-β) [40]. They suppress the immune response in the functional homeostasis of the immune system [35]. Tregs induce immune tolerance by modulating antigen presentation by expressing soluble anti-inflammatory mediators, such as IL-10 and TGFβ, the consumption of IL-2, and the expression of negative regulatory cell surface receptors, including CTLA-4 [41,42]. Patients with MM have elevated Treg level, which is a marker of poor prognosis [43,44,45]. MM cells secrete an inducible T-cell co-stimulator ligand (ICOS-L) and transform non-Tregs into Tregs [46]. Thus, decreasing the Treg count can enhance the immune activity against myeloma cells.

Data on the kinetics of B cells are limited compared with those of T cells. Decreasing levels of polyclonal immunoglobulin reflect suppression of CD19^+^ B cells, which is inversely correlated with disease progression and affects normal B-cell differentiation [21,47]. TGF-β contributes to B-cell dysfunction in myeloma [21,48]. Regulatory B cells (Breg), a small B-cell subset, regulate immune responses via stimulation of IL-10, an anti-inflammatory cytokine, and modulation of CD4^+^ T cell activation and differentiation [49]. Breg induces an immunosuppressive BM microenvironment, which may, in turn, affect therapeutic response and disease outcome in patients with MM [50]. Therefore, Breg inhibition is a potential therapeutic target.

Macrophages are blood cells derived from monocytes and show various activities depending on the body site. Macrophages contribute to antibody-dependent cellular phagocytosis (ADCP) activity, which is the phagocytosis of antibody-opsonized tumor cells via binding to Fcγ receptors present on macrophages or monocytes [51]. In contrast, some macrophages suppress immune activity in myeloma. Tumor-associated macrophages (TAMs) are categorized as M2 and have a pro-tumoral function. They infiltrate the tumors and are associated with the growth, angiogenesis, and metastasis of various cancers, including MM [52,53]. TAMs in MM have little cytotoxicity and suppress T cell activity [54]. In addition, TAMs regulate fibroblast function in BM [52] and induce resistance to chemotherapy via inhibition of Bcl-XL-dependent caspase activation [55]. Thus, TAM inhibition is a potential therapeutic target as well. A summary of function of immune cells for healthy individuals and MM patients is shown in Table 1.

## 3. Importance of Immunological Environment for Long-Term Survival in MM

Immune reconstitution, which is indicated by lymphocyte count in PB and immunoglobulin levels, predicts good prognosis in patients with MM. Changes in the BM immune microenvironment have recently been analyzed using cytometry by time of flight (CyTOF) and NGF.

Early immune reconstitution, which is defined as recovery to the normal count of both lymphocytes and monocytes one month after treatment, predicts long OS. The OS in patients with early immune reconstitution was reported to be similar to that in patients without immune deregulation at diagnosis. The frequency of early immune reconstitution is high in patients treated with IMiDs and low in patients with HRCA [76]. The recovery of absolute lymphocyte count (ALC), defined as ≥ 1400 cells/μL at day 0, day 15, and day 90 after ASCT, predict long OS [15]. Decreased CD4^+^ T cell count and the CD4/8 ratio are associated with poor prognosis [23,77]. The number of clonal CD8^+^ T cells, which are identified as effector memory T cells with a restricted T-cell receptor (TCR) Vβ expression, was associated with persistent stimulation by myeloma-associated antigens [78]. The count of clonal CD8^+^ T cells in PB was higher in myeloma patients who survived for more than 10 years than in those who died in less than 10 years [79]. Among patients with long-term CR after ASCT, the distribution of CD4^+^ and CD8^+^ memory T cells and naïve B cells in PB was higher than that in age-matched healthy individuals [80]. In contrast, the presence of naïve and terminally differentiated T cells in the BM predicted a short survival time in myeloma patients who received ASCT using CyTOF [81]. These results suggested that naïve T cells could not activate antigen engagement, and terminally differentiated T cells could not mediate effective clearance of myeloma cells because of T cell exhaustion. In the transplant-ineligible patients enrolled in the PETHEMA/GEM2010MAS65 study, PFS and OS were longer in the groups rich in naïve and memory B cells in BM using NGF instead of the ISS and those with cytogenetic abnormality and MRD status [82]. Thus, improving the immune microenvironment is associated with long-term survival and is a surrogate marker for good outcomes in myeloma patients.

Immunoparesis is associated with poor outcomes in patients treated with novel agents and cytotoxic agents [83,84,85]. In a multivariate analysis including age, ISS stage, and genetic risk of MM patients treated with novel agents, a low immunoglobulin (Ig)M level was a significant predictor of short PFS and OS compared with IgA and IgG levels [83].

The oligoclonal band is often identified in patients with CR and is an immunoglobulin derived from myeloma cells and other polyclonal B cells [86,87]. In a previous study, the oligoclonal band predicted longer survival and disappeared before relapse [86]. However, in MM patients with extramedullary disease (EMD) or light chain escape, the oligoclonal band may remain. Thus, the oligoclonal band is also considered a form of immune reconstitution [88]. Oligoclonal bands, identified as clonal isotype switches, are often detected after ASCT or chemotherapy [89,90] and predict long-term survival in patients receiving ASCT [89,91]. In patients treated with PI or IMiDs induction therapy followed by ASCT, the frequency of oligoclonal bands in the patients treated with IMiDs was higher than in those without IMiDs. The CD8^+^ T cell count was significantly lower, and the CD4/CD8 T cell ratio was significantly higher in patients with oligoclonal bands than in those without oligoclonal bands [89]. Thus, improvement of immune status concerning T cells and B cells predicts good outcomes in MM patients. In addition, immune reconstitution of T cells could be correlated with that of B cells.

Immune status in BM was similar among healthy individuals, MGUS patients, and MM patients treated with ASCT using CyTOF [92]. Thus, the immune environment in active MM patients is suitable for the proliferation and survival of myeloma cells, while the immune environment in MM patients with remission is unfit for myeloma cells and normal immune cells increases, leading to immune reconstitution.

## 4. Treatment to Improve the Immunological Environment

Currently, IMiDs, PIs, and MoAbs play an important role in MM treatment. Many studies have demonstrated that IMiDs and MoAbs improve the immunological environment via activation of T and NK cells and suppression of Tregs. ASCT improves the immunological environment via the supply of autografts and high-dose melphalan (HD-MEL). In addition, PIs activate ADCC via the downregulation of human leukocyte antigen (HLA) on myeloma cells. A summary of anti-myeloma agents for immune system is shown in Table 1.

### 4.1. IMiDs

IMiDs induce immune modulation and exhibit anti-myeloma activity. They also enhance both the adaptive and innate immune systems via co-stimulation of T cells and enhancement of NK in vitro [56]. These drugs enhance the tumor-specific Th1 type immune response via the generation of IFN-gamma and IL-2 derived from CD4^+^ and CD8^+^ T cells [56]. Examples of IMiDs include lenalidomide (LEN) and pomalidomide (POM), and thalidomide (THAL). LEN and POM, excluding THAL, augment ADCC and innate cytotoxicity of NK cells and promote the proliferation of NK cells dependent on IL-2 [58]. LEN and POM upregulate the expression of FasL and Granzyme B on NK cells [59]. In addition, IMiDs suppress Treg function by downregulating Foxp3 gene expression in Tregs [60]. The effect of IMiDs depends on cereblon (CRBN) burden concerning both direct antitumor effects and immune activity [93]. IMiDs also act on immune checkpoint molecules to enhance immune responses. They suppress PD-1 expression on T and NK cells [57], and LEN suppresses PD-L1 expression in MM cells [43]. Finally, LEN coverts M2 macrophages, identified as TAM, into M1 macrophages via the degradation of the IKAROS family zinc finger 1 (IKZF1) of macrophages [61]. Iberdomide (IBER), a next-generation cereblon-targeting agent, showed direct anti-myeloma activity for LEN- and/or POM-resistant human myeloma cell lines via faster degradation of IKZF1 due to high cereblon-binding affinity and enhanced immunomodulatory effect in a co-culture with peripheral blood mononuclear cells via elevated IL-2 secretion and granzyme-B degranulation [94,95]. In addition, IBER combined with daratumumab (DARA) enhanced complement-dependent cytotoxicity (CDC) compared with either drug alone [95], demonstrating the clinical efficacy of IMiDs and/or anti-CD38 MoAb in refractory MM patients [96].

### 4.2. MoAbs

Daratumumab (DARA) is a monoclonal antibody against CD38 antigen with complement-dependent cytotoxicity (CDC) and enhanced ADCC activities [63]. CD38 is expressed on the surface of myeloma cells as well as T and NK cells [62]. Notably, DARA reduces NK cell count [64]. However, CD38^−/low^ NK cells play a major role in ADCC activity, an immune response that is expected even after DARA administration [65]. DARA also contributes to a clonal increase in CTL and suppression of Treg and Breg [62]. DARA-containing treatment is an option for post-transplant treatment as it might reduce the number of CD38^+^ Treg, which increases early after ASCT and suppresses the immune effect [62]. In the POLLUX trial, the clonal expansion of CD8^+^ T cell and reduction of CD38^+^ Treg were observed more frequently in the deep responders treated with DRd (DARA, LEN, plus dexamethasone (DEX)) than in those treated with Rd (LEN plus DEX) [97]. Anti-CD38 MoAbs also activate T cell function via suppression of adenosine (ADO) production because CD38 functions as an ectoenzyme and promotes ADO production from NAD^+^ via regulation of calcium signaling, which suppresses T cell activity [66,67,68]. Isatuximab (ISA), a new anti-CD38 MoAb, has been approved for relapsed/refractory MM (RRMM). Similar to DARA, ISA can also modulate the immune system. ISA has a higher direct killing activity of myeloma cells [98] but lower CDC activity than DARA [99]. A combination of ISA and POM showed higher anti-myeloma activity than a combination of ISA and LEN [98].

Anti-PD-1 MoAb is expected to be active against several solid malignancies because PD-1-expressing lymphocytes expand tumor-specific CTLs and suppress the anti-myeloma activity [31]. In the KEYNOTE-185 trial, PFS of patients in the pembrolizumab, an anti-PD-1 MoAb, plus Rd group was similar to that of patients in the Rd group, while severe adverse events and treatment-related mortality of patients in the pembrolizumab plus Rd group were higher than those of patients in the Rd group [100].

CD38/PD-1 double-positive lymphocytes have been identified in patients with malignancies [32]. CD38 expression on T cells might induce escape from PD-1/PD-L1 blockade in tumor cells [33]. However, in a previous study, durvalumab, an anti-PD-L1 MoAb, and daratumumab were ineffective in myeloma patients with daratumumab refractoriness [101]. A clinical trial on combined anti CD38 and PD-1 monoclonal antibodies for myeloma patients is ongoing.

Belantamab mafodotin is the first anti-B-cell mature antigen (BCMA) antibody drug conjugate (ADC) with mono methyl auristatin F (MMAF) and has been approved for RRMM in the United States [102]. Belantamab mafodotin not only delivers MMAF into the BCMA-expressed myeloma cells and induces apoptosis but also enhances ADCC and ADCP [70].

### 4.3. ASCT

The therapeutic effect of ASCT depends on HD-MEL therapy for MM. However, autografts play an important role in the therapeutic effect of ASCT. Granulocytes, platelets, and red blood cells recover within a few weeks, while the recovery of other blood cells, such as lymphocytes and monocytes, takes a longer duration after HSC transplantation [103]. Immune reconstitution, which has been shown to increase absolute lymphocyte count, CD4/CD8 ratio, and oligoclonal band, is associated with good clinical outcomes in patients receiving ASCT [15,77,79,80,81,89].

HD-MEL is the mainstay of pre-transplantation conditioning chemotherapy. HD-MEL induces lymphodepletion, which could affect immunological activity, including that of T cells, in myeloma cells [71]. However, T cell count from autografts is upregulated by IL-7 and IL-15, whose serum levels increase after HD-MEL followed by ASCT, although T cell count decreases in the absence of these cytokines [71]. Melphalan has been shown to activate CD8^+^ T cells via dendritic cell activation due to immunogenic cell death and antigen presentation, including the release of high-mobility group box 1 (HMGB-1), in myeloma-bearing mice [72]. In addition, the combination of melphalan and CD4^+^ T cell adoptive cell therapy is more effective than either treatment alone in mice. HD-MEL can enhance immune activity by reducing the levels of Treg and myeloid-derived suppressor cells, which inhibit the anti-myeloma T cell-mediated immune response in BM [72,73].

### 4.4. Proteasome Inhibitors

PIs enhance ADCC activity by suppressing HLA class 1 expression on MM cells [74]. Recently, bortezomib, a proteasome inhibitor, enhanced the ADCC of DARA in mice when human myeloma cell lines and ex vivo NK cells were co-cultured [75]. Additionally, PIs activate dendritic cells by increasing exposure to tumor antigens, thus inducing immunogenic cell death [104,105].

## 5. Clinical Significance of MRD Negativity in MM

MRD can be analyzed in patients who have achieved CR using NGF or NGS. Immunophenotypic and molecular CR are defined as achievement of MRD negativity using NGF and NGS, respectively, according to the International Myeloma Working Group recommendation [106]. Several reports indicate that achievement of negative MRD is associated with prolonged PFS and OS, but the curability of MM is still debatable [107,108]. According to a recent meta-analysis, MRD negativity predicted long PFS and OS independent of transplant eligibility, disease status, cytogenetic risk, MRD sensitivity threshold, and MRD detection methods [5]. A summary of phase 3 clinical trials investigating MRD status was shown in Table 2.

In a phase 2 trial on LEN maintenance therapy after ASCT, no relapse was noted in patients with sustained MRD negativity for two years after a median follow-up time of 40.7 months, while the PFS in patients with loss of MRD negativity was shorter than in those with persistent MRD positivity [16]. MRD status might affect decisions regarding treatment discontinuation or escalation/de-escalation of treatment intensity [117], and several response-adapted clinical trials on MRD are ongoing [118]. Thus, MRD negativity is an important prognostic factor for long-term survival and a biomarker of the treatment strategy.

MRD status is assessed using BM samples of myeloma patients, but the invasiveness of BM tests has necessitated the use of PB samples [119,120,121]. Although MRD assessment using PB samples is less invasive, it is less accurate than using BM samples. Nevertheless, MDS-negativity using PB samples can predict survival in myeloma patients [119,120]. Meanwhile, MRD status using PB might reflect not only residual myeloma cells in the BM but also EMD due to the detection of circulating residual myeloma cells [121]. Therefore, we considered that the use of PB samples is suitable for checking MRD-negativity, while the use of BM samples is necessary for evaluating precise MRD-negativity. Sustained MRD-negativity is important for long-term survival among myeloma patients, but it is controversial whether BM and PB samples are suitable for assessing sustained MRD-negativity. There is no consensus about the optimal timing of the next treatment for patients whose MRD status has changed from negative to positive. We consider that PB samples may be more suitable than BM samples for analysis of sustained MRD-negativity because MRD status using PB reflects residual myeloma cells in not only testing site of BM but also the other sites including EMD.

## 6. Current Treatment to Achieve Persistent MRD-Negativity

High-dose chemotherapy followed by ASCT was developed in the 1990s and is still a standard treatment for patients with MM. In addition, various treatment agents, including IMiDs, PIs, and MoAbs, have been developed in the last two decades, thus increasing CR ratios [1,2]. Many clinical trials have demonstrated that the administration of several treatment agents with different modes of action helps achieve MRD negativity in addition to CR [5,7,8,111,112,114,115,122,123,124]. For MM patients who achieve MRD negativity after induction therapy and ASCT, consolidation and/or maintenance therapy is needed to enhance and maintain the therapeutic effect.

Combination chemotherapies with PIs and IMiDs might induce MRD negativity more frequently than PIs or IMiDs alone for newly diagnosed multiple myeloma (NDMM) and RRMM, although there was no direct comparison between them [122,123]. ASCT and DARA play an important role in achieving MRD negativity after administration of PIs and IMiDs combination chemotherapies [7,14,111,112,114,115,124]. However, no clinical trials have compared up-front ASCT and DARA-containing treatments for NDMM. Considering the improvement of the immune environment after ASCT [125], IMiDs should be one of the best treatment choices. According to a meta-analysis, LEN maintenance therapy prolonged PFS and OS [126,127]. In addition, LEN maintenance therapy achieved and maintained MRD negativity compared with no maintenance therapy [128]. In the TOURMALINE-MM3 trial, post-transplantation ixazomib maintenance therapy prolonged PFS and prevented conversion from MRD negative to positive compared with placebo in MRD-negative patients before maintenance therapy [129]. Moreover, use of MoAbs for consolidation and/or maintenance therapies has been studied [13,112,130]. However, in the CASSIOPEIA trial, the clinical significance of DARA maintenance therapy was relatively low in patients treated with D-VTD (DARA, BOR, THAL, and DEX) therapy as induction therapy compared with those treated with VTD therapy [131]. Thus, further studies are needed on DARA maintenance therapy regarding the duration and timing of discontinuation.

## 7. Characteristics of Residual MM Cells in MRD Positive Patients

Combination treatment of several agents with different modes of action, including ASCT, contributes to the achievement of MRD negativity. Some clinical trials concerning MRD status-adapted treatment strategies are currently ongoing (Table 3). The MASTER trial investigated the efficacy of four cycles of D-KRd (DARA, CFZ, LEN, plus DEX) followed by ASCT and eight cycles of D-KRd consolidation treatment in achieving MRD negativity for NDMM [132]. If MRD negativity was achieved, chemotherapy was discontinued. Approximately 62–78% of patients achieved MRD negativity after the treatment, including eight cycles of consolidation therapy, and treatment could be discontinued. However, some patients do not achieve MRD negativity due to the presence of residual myeloma cells that are resistant to the administered treatment. Such patients might require a change of treatment strategy; however, to our best knowledge, no MRD status-adapted treatment strategy has been established. In addition, the clinical significance of pre-emptive therapy for RRMM with conversion from MRD negativity to positivity has not been analyzed to date. The PREDATOR trial is investigating the efficacy of pre-emptive therapy for patients with RRMM who have achieved MRD negativity after the last chemotherapy and may reveal the improvement of survival time of patients with RRMM whose MRD status converted to positivity [NCT03697655]. Therefore, it is important to understand the characteristics of residual myeloma cells and select suitable treatment strategies. Previous reports have demonstrated that residual myeloma cells are immature, have low CD38 expression, and are rich in integrin-related antigens [133,134]. Thus, it has been suggested that residual myeloma cells may be resistant to ongoing chemotherapy through adhesion to bone marrow stromal cells (BMSCs) and have some advantages, such as growth, proliferation, survival, and resistance to chemotherapy [134,135]. In addition, a previous study reported that when the residual myeloma cells among patients who received VMP (BOR, melphalan, and prednisone) and alternate treatment with VMP and Rd were analyzed by flow cytometry, the incidence of surface antigen changes from the time of diagnosis to MRD assessment in the VMP alone group was higher than in the alternative treatment group [134]. These data suggest that the use of several therapeutic agents might reduce acquired chemotherapy resistance.

## 8. Drug Resistance concerning Gene Mutation

To date, few reports have examined the genetic characteristics of residual myeloma cells [136]. Possibly, the residual myeloma cells in MRD-positive patients on LEN maintenance therapy are resistant to LEN due to decreased CRBN burden, *CRBN* gene mutation, and *c-MYC* overexpression [136,137,138,139]. IMiDs may not be effective against residual myeloma cells with reduced CRBN burden and *CBRN* gene mutations. In these cases, chemotherapy with a different mechanism of action, such as PI or MoAb, should be selected. PI is a therapeutic option for residual myeloma cells with refractoriness for LEN via *c-MYC* overexpression because PI has been reported to have a therapeutic effect in patients with *c-MYC* overexpression [140,141,142]. MRD positivity during PI maintenance therapy means that the residual myeloma cells may be resistant to PIs. One of the causes of refractoriness to PIs is proteasome 20S subunit beta 5 (*PSMB5*) gene mutations [136,143]. Notably, a previous study found that the frequencies of gene mutations concerning refractoriness to LEN or PIs were low, suggesting that these mutations were detected in subclonal myeloma cells, and the clinical significance of these mutations, including gene mutation-guided treatment strategy, should be analyzed in future studies [144].

The causes of resistance to anti-CD38 MoAbs are categorized into decreased or loss of CD38 expression, neutralization of CD38, and decreased immunological effects, including ADCC, CDC, and ADCP, via reduced numbers of NK cells, complement inhibitory proteins, and CD47 expression on myeloma cells, respectively [145]. CD38 expression decreased in myeloma cells just after DARA was administered and recovered approximately six months after DARA was discontinued [146]. In a phase 2 trial of ISA monotherapy in the RRMM with refractoriness to DARA, a high CD38 expression was associated with a long interval from the last DARA administration, and the disease control rate was higher in the six months or longer DARA-free interval group than in the three months or shorter DARA-free interval group [147]. In the ICARIA-MM trial, the PFS of DARA as the first subsequent therapy in the ISA and POM plus DEX (Pd) group was shorter than that in the Pd group [148]. Thus, the efficacy of anti-CD38 MoAb could be related to the anti-CD38 free interval, and the immunophenotypic characteristics of residual myeloma cells could predict resistance to anti-CD38 MoAbs.

## 9. Immunological Treatment to Eradicate Residual MM Cells

Residual myeloma cells acquire drug resistance via gene mutation and immune escape. Thus, an immunological approach can be essential for eradicating residual myeloma cells. The commonly used immunotherapy is allogeneic hematopoietic stem cell transplantation (HSCT) up to a few years ago. Treatment of young high-risk myeloma patients with allogeneic HSCT using reduced-intensity conditioning treatment after up-front ASCT has been investigated for several decades, but there is no consensus on whether these treatment strategies can prolong survival [149]. Two clinical trials have revealed long survival of patients treated with ASCT followed by allogeneic HSCT compared to those treated with tandem ASCT [150,151]. In contrast, a clinical trial on allogeneic HSCT upfront ASCT reported that OS in patients who received allogeneic HSCT was similar to that in patients who received ASCT alone because some patients experienced recurrence and transplant-related mortality after allogeneic transplantation [152]. Therefore, other immunotherapies are required. Since PD-L1 is highly expressed in residual myeloma cells in MRD-positive patients [153], the inhibition of immune checkpoints via blocking PD1/PDL1 is expected. Although several reports of post-ASCT treatment with pembrolizumab have shown tolerability, no therapeutic effect or survival benefit has been demonstrated [154,155].

Chimeric antigen receptor T cell (CAR-T) is a new treatment option as immunotherapy and is currently approved by the US FDA (Food and Drug Administration) for patients with RRMM. Several clinical trials are investigating the efficacy and tolerability of CAR-T as an earlier line of treatment such as consolidation therapy [156]. The most popular target antigen for CAR-T is the BCMA, which is specifically expressed in myeloma cells [156,157,158,159]. The clinical outcome of CAR-T cells is associated with the quality of harvested autologous T-cells [160,161]. In a clinical trial for RRMM, anti-BCMA CAR-T expansion and response were related to the preserved CD4/8 ratio and high levels of naïve and stem cell memory T cells during mobilization and CAR-T manufacturing [145,146]. The T-cells suitable for CAR-T were identified more frequently in early-phase MM patients than in heavily treated MM patients [162]. Thus, post-ASCT is one of the best timings for immune therapy, including CAR-T.

Bispecific T-cell engager (BiTE) is an important immunotherapy option. Generally, BiTE is more tolerable than CAR-T, considering the incidence of cytokine release syndrome and neurological toxicity, and less effective than CAR-T according to the results of clinical trials, although no clinical trial has directly compared the two therapies [163]. BiTE might be suitable for MRD-positive patients after ASCT because the immune environment is improved by the autograft. The excellent therapeutic effect of anti-BCMA CAR-T is clear, as evidenced by the high MRD negativity rate, but the PFS is not as long as expected [158]. The resistance mechanisms to anti-BCMA CAR-T include immune escape through reduced expression or disappearance of BCMA on the surface of myeloma cells [164,165]. To maintain MRD negativity, continuous stimulation of CAR-T cells could be important. Considering this suggestion, clinical trials on the addition of IMiDs after CAR-T are under way [166]. It has also been reported that the use of BiTE after CAR-T treatment stimulates CAR-T cells and reactivates the immune response. In addition, anti-CD19 and BCMA dual CAR-T was developed to overcome the resistance due to exhaustion of CAR-T cells [167]. In the future, it may be necessary to develop a treatment strategy for maintenance of immune activation with IMiDs, MoAb, or BiTE after CART for the purpose of maintaining MRD negativity.

## 10. Correlation between MRD Status and Immune Environment

Both the immunological environment and MRD-negativity are essential for long-term survival in patients with MM. The immunological environment may be associated with the MRD status [168]. Previous studies reported that the number of TAM, erythroblasts, Tregs, memory B cells, and CD4^+^ T cells (especially CD27^+^) in BM of MRD-positive patients were significantly higher than those in BM of MRD-negative patients [18,82]. Notably, the number of immune cells in PB does not reflect that in BM [18]. Effector Treg, a form of Treg that strongly suppresses immune activity in myeloma cells, exists in the BM tumor site, although there is no significant difference in Treg counts in PB and BM [168,169]. In patients who received ASCT followed by LEN maintenance therapy, NK cells decreased, and exhausted T cells increased in PB of MRD-positive patients compared with that of MRD-negative patients [170]. In addition, the expression of killer cell immunoglobulin like receptor, 2 Ig domains and short cytoplasmic tail 4 (KIR2DS4), which activates immunity, decreased, and that of NKG2A, which suppresses immunity, increased in the MRD-negative patients compared with the MRD-positive patients [170]. Thus, improving the immune environment can contribute to the achievement of MRD, and eradicating residual myeloma cells can balance the immune environment. The correlation between disease status, including MRD status, and immunological environment is shown in Figure 1.

## 11. Future Directions

IMiDs, PIs, MoAbs, and ASCT are important for improving the immune environment and maintaining MRD negativity. We previously reported that the total therapy approach combining these four treatment approaches could be essential, considering the BM microenvironment, such as adhesion to BMSCs, vascular niche, and endosteal niche [171]. If MRD-negativity is achieved, current treatment should continue, although the timing of treatment discontinuation is still controversial. However, immune-activating agents, such as IMiDs, anti-CD38 MoAb, or anti-PD-1 MoAb, might be suitable because the immune environment is equivalent in patients with MRD negativity. If MRD status is positive, the characteristics of residual myeloma cells, such as genetics and immunophenotypes, should be analyzed to optimize treatment. In particular, we considered that assessing MRD status using PB samples might be more suitable for optimizing treatment than using BM samples because MRD status using PB samples can reflect both residual myeloma cells in testing site of BM and the other sites, including EMD [121]. Besides, the use of PB sample eases analysis of residual myeloma cells, and the immunological microenvironment might be activated compared with those in patients with active disease or MRD negativity. Treatment algorithm concerning MRD and immunological status for NDMM when anti-CD38 MoAb, IMiDs, and PIs are available was shown in Figure 2.

If anti-BCMA CAR-T is available, these agents may be reasonable considering their different modes of action. Meanwhile, immune-activating agents, except anti-PD-1 MoAb, may not be effective because the immune environment is still activated in patients with MRD positivity. Blockade of PD-1/PD-L1 may be necessary as residual myeloma cells express PD-L1. The treatment strategies considering the MRD status are shown in Figure 3. However, analysis and characterization of residual myeloma cells are currently difficult in most hospitals, underscoring the need to consider the resistance to the current anti-myeloma agents. For example, if MRD status is positive during LEN maintenance therapy, the treatment can be changed, including class switching, considering decreased CBRN burden, *CBRN* mutation, and/or *c-MYC* upregulation. However, it has also been argued when the treatment should be changed during persistent MRD-positivity considering the possibility of a late responder to current treatment. Data from clinical trials on the current MRD-driven treatment will provide more insights into the effective treatment approaches [172].

Examining the improvement of the immune environment is difficult in current practice, but it can be predicted using recently reported indicators, such as the lymphocyte to monocyte ratio (LMR) [173,174,175]. A high LMR status reflects a good immunological environment and is associated with a long survival time among MM patients. Recently, we demonstrated that PFS in patients with both MRD-positivity and low LMR status was significantly shorter than in those with MRD-negativity and/or high LMR status, despite the achievement of CR [176]. Thus, a treatment change might be considered in patients with both MRD positivity and low LMR status. However, there is no current evidence showing the clinical significance of changing treatment approaches to enhance the treatment response and improve the immune environment.

## 12. Conclusions

We consider that improvement of the immune environment and maintenance of MRD negativity are key factors for the long-term survival of MM patients. Considering the microenvironment around myeloma cells, initial treatment encompassing IMiDs, PIs, anti-CD38 MoAb, and ASCT is important. This total therapy approach can improve the immune environment and help achieve MRD negativity. This review suggests that an MRD-driven treatment strategy may be promising, but genetic and immunophenotypic analyses of residual myeloma cells should be repeated to select a suitable treatment for residual myeloma cells. Before these analyses are available in clinical practice, treatment can be selected based on “class switch.” In the future, there is a need to develop a treatment strategy that not only treats the myeloma cells but also improves the immune environment and targets the residual myeloma cells.

## Figures and Tables

**Figure 1 cancers-13-04867-f001:**
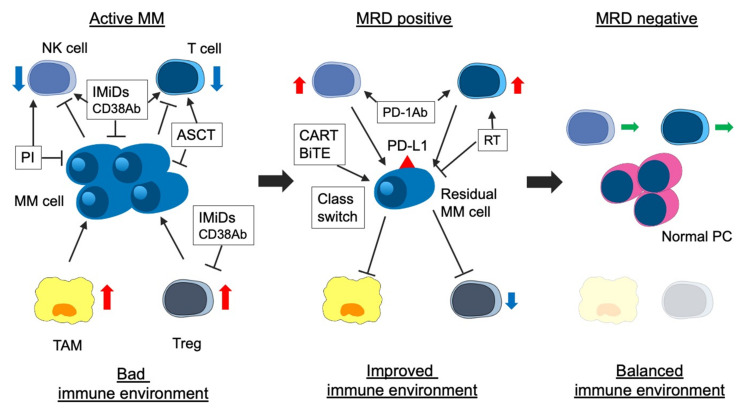
Association between disease status and immune environment in myeloma. The immune environment contributes to immune escape, proliferation, and survival of myeloma cells via several cytokines and activated immunosuppressed cellular components (i.e., TAM and Treg) in patients with active myeloma. IMiDs, PIs, anti-CD38 MoABs, and ASCT improve the immune environment. ASCT contributes to immune reconstitution. In patients with MRD positivity, the immune environment is improving, but immunosuppressive cells are still active. In addition, residual myeloma cells express immune checkpoint molecules (i.e., PD-L1) and escape from immune attack. Two treatment strategies for residual myeloma cells are considered: the first is the activation of immune response using agents with another mode of action (i.e., anti-PD-1 MoAb). The second is a change into a treatment approach suitable for residual myeloma cells (class switch) or a new mode of action agents (i.e., BCMA-targeting CAR-T). Finally, in patients with MRD-negativity, immune-suppressive cells decrease, the activity of immune cells (CTL and NK cells) is equivalent, and immune reconstitution occurs (balanced immune environment). MM, multiple myeloma; MRD, minimal residual disease; IMiDs, immunomodulatory drugs; PIs, proteasome inhibitors; MoAbs, monoclonal antibodies; ASCT, autologous stem cell transplantation; NK cell, natural killer cell; TAM, tumor associated macrophage; Treg, regulatory T cell; PD-1, progress-death 1; PD-L1 progress–death ligand 1; BCMA, B-cell mature antigen; and CAR-T, chimeric antigen receptor T cell.

**Figure 2 cancers-13-04867-f002:**
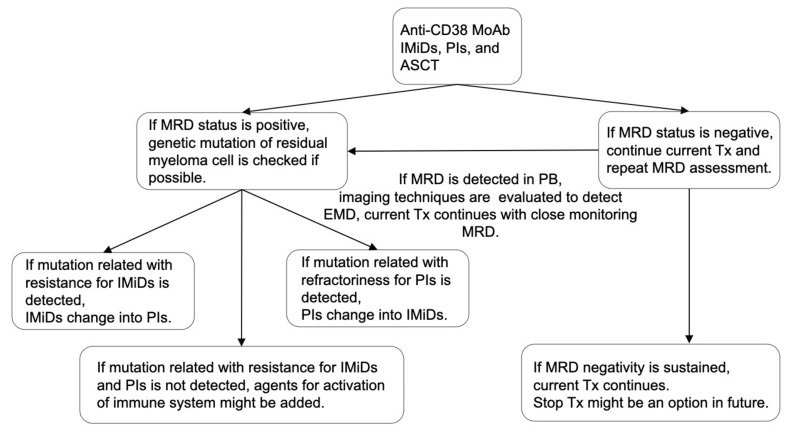
Treatment algorithm concerning MRD and immunological status. Combination therapy with anti-CD38 MoAb, IMiDs, PIs, and ASCT may be suitable for MM patients considering the efficacy against myeloma cells and improved immune system. If the MRD status is negative, the current treatment should continue. However, if the MRD status is positive, the genetic mutation of residual myeloma cells should be analyzed to optimize treatment. If mutation related with resistance for IMiDs or PIs is detected, treatment should be changed. If mutation related with resistance for IMiDs or PIs is not detected, agents for activation of immune system might be added. MRD assessment should be repeated in the patients with MRD negativity. If MRD status convert into positivity in PB sample, imaging technique should be evaluated to detect EMD. Thereafter, the current Tx could continue with close monitoring MRD status. MRD, minimal residual disease; IMiDs, immunomodulatory drugs; PIs, proteasome inhibitors; MoAbs, monoclonal antibodies; ASCT, autologous stem cell transplantation; Tx, treatment; PB, peripheral blood; and EMD, extramedullary disease.

**Figure 3 cancers-13-04867-f003:**
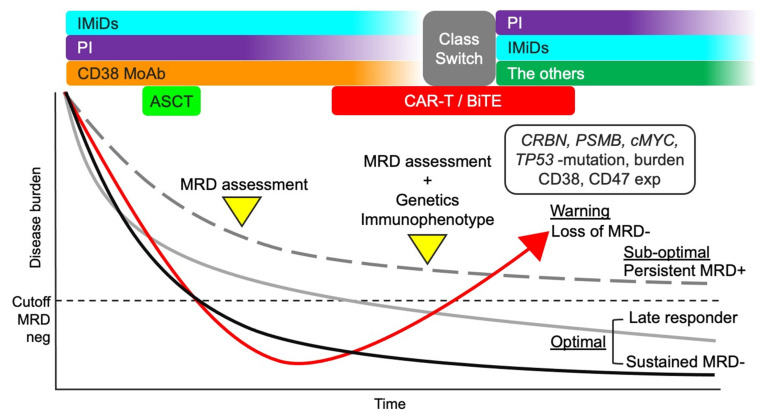
Treatment strategy considering MRD status. A total therapy approach combining IMiDs, PIs, anti-CD38 MoAb, and ASCT may be suitable for MM patients considering the efficacy against myeloma cells and improved microenvironment. MRD status after the total therapy approach can be useful in further treatment decisions. If the MRD status is negative, the current treatment should continue (optimal). However, if the MRD status is positive, the genetic and immunophenotypic characteristics of residual myeloma cells should be analyzed to optimize treatment. Loss of MRD negativity can lead to aggressive recurrence (warning). The clinical outcome of persistent MRD positivity is better than that of loss of MRD-negativity (sub-optimal). Repeated MRD assessment may be necessary for patients with persistent MRD positivity to identify late responders and detect early-phase recurrence. MM, multiple myeloma; MRD, minimal residual disease; IMiDs, immunomodulatory drugs; PIs, proteasome inhibitors; MoAbs, monoclonal antibodies; ASCT, autologous stem cell transplantation; CAR-T, chimeric antigen receptor T cell; CRBN, cereblon; PSMB5, proteasome 20S subunit beta 5; and exp, expression.

**Table 1 cancers-13-04867-t001:** Summary of function of immune cells and anti-myeloma agents for immune system. IMiDs, immunomodulatory drugs; MoAb, monoclonal antibody; PD-1, programed death-1; HD-MEL, high dose-melphalan; ASCT, autologous stem cell transplantation; PIs, proteasome inhibitor; CTL, cytotoxic T-cell; NK cell, natural killer cell; Treg, regulatory T-cell; Breg, regulatory B cell; TAM, tumor associated macrophage; MM, multiple myeloma; IL-2, interleukine-2; IFN-gamma, interferon gamma; ICOS-L, inducible T-cell co-stimulator ligand; ADCC, antibody-dependent cellular cytotoxicity; ADCP, antibody-dependent cellular phagocytosis; HMGB-1, high-mobility group box-1; FOXP3, forkhead box P3; and ADO, adenosine.

Characteristics	CTL	NK Cell	Treg	Breg	Macrophage	TAM
Function in healthy individuals	Adoptive immune responses activated by IL-2 and anti-tumor effect by releasing IFN-gamma [19]	Innate immune response, and regulate the activities of T cells, macrophages, and dendritic cells [34]	Suppression for immune response via immune tolerance by modulating antigen presentation, the consumption of IL-2, and the expression of negative regulatory cell surface receptors [35,41,42]	Regulation of immune responses via stimulation of IL-10 [49]	Various activities depending on the body site	-
Function in MM patients	Exhaustion in tumor site via upregulation of inhibitory receptors [22,24,26]	NK cell activity is reduced in patients with MM [35]	Elevated Treg level predicts poor prognosis [43,44,45]; transformation from non-Tregs into Tregs by secretion of ICOS-L [46]	Induction of an immunosuppressive BM micro-environment [50]	Relation with ADPC [51]	Suppress T-cell activity [54]. Induction of resistance to chemotherapy [55]
Drugs	-	-	-	-	-	-
IMiDs	Activate via INF-gamma and IL-2 from T-cell from CD4^+^ T-cell and PD-1 blockade [56,57]	Activate ADCC by INF-gamma and IL-2 from CD4^+^ T-cell [56,58,59]	Inhibit via down regulation of FOXP3 [60]	-	-	Decrease TAM via conversion from TAM into M1 macrophage [61]
Anti-CD38 MoAb	Induce clonal increasing CTL [62]	Decrease CD38^+^ NK cell but activate ADCC by CD38^−/low^ NK cell [63,64,65]	Inhibit activity via suppression ADO [66,67,68]	Inhibit activity via suppression ADO [66,67,68]	Activate ADCP [69]	-
Anti-PD-1 MoAb	Activate via PD-1 blockade [31]	Activate via PD-1 blockade [31]	-	-	-	-
Belantamab Mafodotin		Activate ADCC [70]	-	-	Activate ADCP [70]	-
HD-MEL + ASCT	Decrease T-cell but activate via activated dendritic cell by releasing HMGB1 [71,72]	-	Decrease Treg by HD-MEL [72,73]	-	-	-
PIs	-	Activate ADCC via HLA class1 blockade [74,75]	-	-	-	-

**Table 2 cancers-13-04867-t002:** Summary of phase 3 clinical trials investigating MRD status. MRD, minimal residual disease; TE-NDMM, transplantation eligible newly diagnosed multiple myeloma; NTE-NDMM, not transplantation eligible newly diagnosed multiple myeloma; RRMM, relapsed or refractory multiple myeloma; NGS, next generation sequencing; MCF, multi-color flowcytometry; DARA, daratumumab; LEN, lenalidomide; K, carfilzomib (CFZ); V, bortezomib (BOR); T, thalidomide (THAL); M, melphalan (MEL); C, cyclophosphamide (CPA); d, dexamethasone (DEX); ASCT, autologous stem cell transplantation; CVd, CPA, BOR, plus DEX; VMP, BOR, MEL, plus prednisone; VRd, BOR, LEN, plus DEX; VTd, BOR, THAL, plus DEX; D-VTd, DARA, plus VTd; KRd, CFZ, LEN, plus DEX; KCd, CFZ, CPA, plus DEX; Rd, LEN, plus DEX; DRd, DARA, plus Rd; VMP, BOR, MEL, plus prednisone; D-VMP; DARA, plus VMP; Vd, BOR, plus DEX; DVd, DARA, plus Vd; Kd, CFZ, plus DEX; DKd, DARA, plus Kd; CONS, consolidation therapy; MT, maintenance therapy; OBS, observation; PFS, progression free survival; OS, overall survival; NR, not reached; and mo, months.

Trial	Disease Status	Cutoff of MRD Negativity (Method)	Treatment	MRD Negative Rate	Outcome (MRD− vs. MRD+)
IFM2009 [108]	TE-NDMM	10^−6^ (NGS)	VRd followed by ASCT followed by VRd vs. VRd alone followed by LEN-MT	ASCT arm 30%, VRd alone arm 20%	Median PFS: NR vs. 29 mo
EMN02/HO95 [109]	TE-NDMM	10^−5^ (MCF)	CVD followed by VMP vs. ASCT followed by VRd as CONS vs. OBS followed by LEN-MT	Post CONS: 76%	5yr PFS: 79% vs. 48%
RPIMeR [110]	TE-NDMM	10^−5^(MCF)	Tandem ASCT vs. single ASCT vs. ASCT + 4cycles VRd	After 1year post first ASCT MRD, tandem ASCT 92%, single ASCT 78%, ASCT + VRd 85%	PFS 76% vs. 44%, OS 96% vs. 66%
FORTE [111]	TE-NDMM	10^−5^ (NGS)	KRd followed by ASCT followed by KRd (A) vs. KRd 12 (B) vs. KCd followed by ASCT followed by KCd (C)	Arm A: 42%, Arm B: 58%,Arm C: 54% using NGS (10^−5^) Arm A: 42%, Arm B: 58%, Arm C: 54% using NGS (10^−6^)	-
CASSIOPEIA [112]	TE-NDMM	10^−5^ (MCF) 10^−6^ (NGS)	D-VTd followed by ASCT followed by D-VTd vs. VTd followed by ASCT followed by VTd	Post IND, D-VTd 35%, VTd 23%; Post CONS, D-VTd 64% vs. VTd 44% using MCF (10^−5^); D-VTd 39% vs. VTd 23% using NGS (10^−6^)	-
CASSIOPEIA2 [113]	TE-NDMM	10^−5^ (MCF) 10^−6^ (NGS)	D-MT vs. OBS	D-MT 6.61%, OBS 55.2% using MCF (10^−5^); D-MT 49.5%, OBS 36.7% using NGS (10^−6^)	-
MAIA [7]	NTE-NDMM	10^−5^ (NGS)	DRd vs. Rd	DRd 24.1%, Rd 7.3%	-
ALCYONE [114]	NTE-NDMM	10^−5^ (NGS)	D-VMP followed by D-MT vs. VMP alone	D-VMP 28%, VMP 7%	-
POLLUX [115]	RRMM	10^−5^ (NGS)	DRd vs. Rd	DRd 32.5%, Rd 6.7%	DRd arm, NR vs. 27.5 mo; Rd arm, 55.3 mo vs. 15.7 mo
CASTOR [115]	RRMM	10^−5^ (NGS)	DVd vs. Vd	DVd 15.1%, Vd 1.6%	DVd arm, NR vs. 12.4 mo; Vd arm, 37.6 mo vs. 6.8 mo
CANDOR [116]	RRMM	10^−5^ (NGS)	DKd vs. Kd	DKd 18%, Kd 4%	

**Table 3 cancers-13-04867-t003:** Clinical trials concerning MRD status adapted treatment strategies. NDMM, newly diagnosed multiple myeloma; RRMM, relapsed or refractory multiple myeloma; MRD, minimal residual disease; PD, progressive disease; Tx, treatment; DARA, daratumumab; LEN, lenalidomide; K, carfilzomib (CFZ); V, bortezomib (BOR); d, dexamethasone (DEX); ASCT, autologous stem cell transplantation; DR, DARA, plus LEN; DRd, DARA, LEN, plus DEX; DKd, DARA, CFZ, plus DEX; D-KRd, DARA, CFZ, LEN, plus DEX; D-VRd, DARA, BOR, LEN, plus DEX; and IMWG, international myeloma working group.

	Phase	Disease Status	Study Design	Primary Endpoint
DRAMMATIC (NCT04071457)	3	NDMM	DR versus LEN alone maintenance after ASCT (randomization1). If MRD+, maintenance Tx continues. If MRD−, maintenance Tx continue versus stop (randomization 2).	Overall survival between DR and LEN alone
REMMANT (NCT04513639)	3	RRMM	Salvage Tx, DKd until PD for the patients with loss MRD− (Arm A) versus PD according to IMWG criteria.	Progression-free survival
MASTER (NCT03224507)	2	NDMM	D-KRd 4 cycles followed by ASCT; consolidation Tx, D-KRd 8 cycles; maintenance Tx, LEN alone until PD. If MRD− achieved after ASCT, 4 or 8 cycles of D-KRd as consolidation Tx, treatment free observation.	MRD− rate at the completion of consolidation Tx
DART4MM (NCT03992170)	2	NDMM	DARA monotherapy every week in 1–8 weeks and every 2 weeks in 9–24 weeks. If MRD+, DARA every 4 weeks for 80 weeks; if MRD−, DARA stop.	Overall response rate
(NCT04140162)	2	NDMM	Induction Tx, DRd 1–24 weeks; consolidation Tx, D-VRD for only MRD+ 25–36weeks; maintenance Tx, DR 37–88 weeks followed by LEN alone until PD.	MRD− rate after induction and consolidation Tx
PREDATOR (NCT03697655)	2	RRMM	Pre-emptive DARA until PD versus observation for MRD+ RRMM after MRD− by last line chemotherapy.	Event-free survival

## Data Availability

No new data were created or analyzed in this study. Data sharing is not applicable to this article.

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
