# Peer review of "Treatment Strategy for Multiple Myeloma to Improve Immunological Environment and Maintain MRD Negativity"

_cancers, 2021, doi:10.3390/cancers13194867_

Round 1

Reviewer 1 Report

This is a well written review about current issues regarding biological and clinical aspects of immune system involvement and MRD in multiple myeloma. I have only a few comments to do. 

1) Check-point related therapies have been currently  abandoned, due to side effects and higher mortality observed in clinical studies. This should be better underlined in the text. 

2) In the IMIDs section, iberdomide could be briefly cited and discussed. 

3) Other antibodies, such as isatuximab, belentamab mafoditin, Bi-specific,  are emerging as possible new treatments for multiple myeloma. They should be at least cited, as well as allogeneic transplantation, as they deeply interact with the immune system. If not, the authors should clarify that these treatments are outside of their interest or, alternatively, that there few data for these approaches  in the context of the present review, if they think so. 

4) Some updates of clinical studies reported at the recent ASCO, EHA and IMW meetings should be implemented.  

Author Response

Reviewer1

This is a well written review about current issues regarding biological and clinical aspects of immune system involvement and MRD in multiple myeloma. I have only a few comments to do. 

  • Check-point related therapies have been currently abandoned, due to side effects and higher mortality observed in clinical studies. This should be better underlined in the text. 

Response: Thank you for pointing out the high mortality due to adverse events of anti PD-1 antibodies. We have emphasized this point in lines 245–248 of the revised manuscript.

In the KEYNOTE-185 trial, PFS of patients in the pembrolizumab plus Rd group was similar to that of patients in the Rd group, while severe adverse events and treatment-related mortality of patients in the pembrolizumab plus Rd group were higher than those of patients in the Rd group [93].

  • In the IMIDs section, iberdomide could be briefly cited and discussed. 

Response: Thank you for pointing out the need to add some discussion on iberdomide. Following your suggestion, we have added a discussion on iberdomide in the revised manuscript, as shown below.

Lines 215–222.

Iberdomide (IBER), a next-generation cereblon-targeting agent, showed direct anti-myeloma activity for LEN- and/or POM-resistant human myeloma cell lines via faster degradation of IKZF1 due to high cereblon-binding affinity and enhanced immunomodulatory effect in a co-culture with peripheral blood mononuclear cells via elevated IL-2 secretion and granzyme-B degranulation [80,81]. In addition, IBER combined with daratumumab (DARA) enhanced complement-dependent cytotoxicity (CDC) compared with either drug alone [81], demonstrating the clinical efficacy of IMiDs and/or anti-CD38 MoAb in refractory MM patients [82].

  • Other antibodies, such as isatuximab, belentamab mafoditin, Bi-specific, are emerging as possible new treatments for multiple myeloma. They should be at least cited, as well as allogeneic transplantation, as they deeply interact with the immune system. If not, the authors should clarify that these treatments are outside of their interest or, alternatively, that there few data for these approaches in the context of the present review, if they think so. 

Response: Thank you for your suggestion about other antibody agents, such as isatuximab, belantamab mafodotin, and bispecific antibody, as well as allogeneic hematopoietic stem cell transplantation. We have added some discussion on these treatments, as shown below.

Lines 237–242.

Isatuximab (ISA), a new anti-CD38 MoAb, has been approved for relapsed/refractory MM (RRMM). Similar to DARA, ISA can also modulate the immune system. ISA has a higher direct killing activity of myeloma cells [91] but lower CDC activity than DARA [92]. A combination of ISA and POM showed higher anti-myeloma activity than a combination of ISA and LEN [91].

Lines 272–276.

Belantamab mafodotin is the first anti-B-cell mature antigen (BCMA) antibody drug conjugate (ADC) with mono methyl auristatin F (MMAF) and has been approved for RRMM in the United States [95]. Belantamab mafodotin not only delivers MMAF into the BCMA-expressed myeloma cells and induces apoptosis but also enhances ADCC and ADCP [96].

Lines 553–568.

Bispecific T-cell engager (BiTE) is an important immunotherapy option. Generally, BiTE is more tolerable than CAR-T, considering the incidence of cytokine release syndrome and neurological toxicity, and less effective than CAR-T according to the results of clinical trials, although no clinical trial has directly compared the two therapies [163]. BiTE might be suitable for MRD+ patients after ASCT because the immune environment is improved by the autograft. The excellent therapeutic effect of anti-BCMA CAR-T is clear, as evidenced by the high MRD negativity rate, but the PFS is not as long as expected [164]. The resistance mechanisms to anti-BCMA CAR-T include immune escape through reduced expression or disappearance of BCMA on the surface of myeloma cells [165, 166]. To maintain MRD negativity, continuous stimulation of CAR-T cells could be important. Considering this suggestion, clinical trials on the addition of IMiDs after CAR-T are underway [167]. It has also been reported that the use of BiTE after CAR-T treatment stimulates CAR-T cells and reactivates the immune response. In addition, anti-CD19 and BCMA dual CAR-T was developed to overcome the resistance due to exhaustion of CAR-T cells [168]. In the future, it may be necessary to develop a treatment strategy for maintenance of immune activation with IMiDs, MoAb, BiTE, etc. after CART for the purpose of maintaining MRD negativity.

We have already mentioned allogeneic hematopoietic stem cell transplantation in section “9. Immunological treatment to eradicate residual MM cells,” as shown below.

Lines 508–518.

The commonly used immunotherapy until a few years ago has been allogeneic hematopoietic stem cell transplantation (HSCT). Treatment of young high-risk myeloma patients with allogeneic HSCT using reduced-intensity conditioning treatment after up-front ASCT has been investigated for several decades, but there is no consensus on whether these treatment strategies can prolong survival [149]. Two clinical trials have revealed long survival of patients treated with ASCT followed by allogeneic HSCT compared to those treated with tandem ASCT [150, 151]. In contrast, a clinical trial on allogeneic HSCT after upfront ASCT reported that OS in patients who received allogeneic HSCT was similar to that in patients who received ASCT alone because some patients experienced recurrence and transplant-related mortality after allogeneic transplantation [152]. Therefore, other immunotherapies are required.

  • Some updates of clinical studies reported at the recent ASCO, EHA and IMW meetings should be implemented.  

Response: Thank you for your suggestion. We have added the latest results of the MASTER trial, as shown below.

Lines 408–413.

The MASTER trial investigated the efficacy of four cycles of D-KRd (DARA, CFZ, LEN plus DEX) followed by ASCT and eight cycles of D-KRd consolidation treatment in achieving MRD negativity for NDMM [132]. If MRD negativity was achieved, chemotherapy was discontinued. Approximately 62–78% of patients achieved MRD negativity after the treatment, including eight cycles of consolidation therapy, and treatment could be discontinued.

Reviewer 2 Report

Major concerns:

     1: This is a review of treatment strategies and yet there are no tables listing these treatments in any of the sections listed. It is very important that tables of studies are generated to understand the sections discussed

Recommend the following:

A: Table of studies looking at T-cells, NK cells, T-regs and macrophages assessment on multiple Myeloma (MM) cells/patients compared to normal- listing the studies, patients #s, treatment use, what measurement was done, what the values are, conclusions etc.

B. Table that shows studies of improved immunological environment with the Imids, proteosome inhibitor(PIs), moabs, and ASCT, compared to control. Layout as in A

C. Table of studies in which MRD assessment was done in Newly dx MM and relapse pts with use of Novel agents (Imids, PIs, Moabs) and list outcomes of MRD neg vs MRD pos.

  1. in section 7: “Characteristics of residual MM cells in MRD positive patients”

The statement “--´However, some patients do not 329 achieve MRD negativity after intensive treatment due to residual myeloma cells that are 330 resistant to the administered agents. Such patients require a change in treatment strategy” cannot be true.

Can you please site studies in which patients with MRD+ after intensive treatment have had a change in their treatment strategies.  As of now, MRD status is not being used to change treatment strategies. For example, a patient in VGPR or CR but MRD+ and on Imid or PI maintenance, are there any studies looking at either changing their maintenance to another regimen vs leaving on len?  Please discuss these studies. As far as I know, there is no published randomized studies on this.  

The same applies for relapse patients who are on a certain regimen. Are their any studies in the relapse setting where MRD status  was used to change treatment vs continue same therapy?  If so please state.

It will also be beneficial for the authors to list ongoing studies looking at assessment of MRD status in newly Diagnosed patients  to determine if stopping maintenance in MRD- pts versus continuation of maintenance made a difference in PFS/OS- for eg the SWOG 1803

  1. Recommend an algorithm figure for MM treatment to improve immunological environment and MRD negativity in newly dx MM pts and in relapse patients

Minor:

1.please make sure font and font size is uniform through out.

  1. in section 9 “ Immunological treatment to eradicate residual MM cells”, the authors forget to site and discuss these studies:

A: “Tandem Autologous-Autologous versus Autologous-Allogeneic Hematopoietic Stem Cell Transplant for Patients with Multiple Myeloma: Long-Term Follow-Up Results from the Blood and Marrow Transplant Clinical Trials Network 0102 Trial”,  by Giralt et al-  BBMT, 2019, 798-804.  That showed Allogeneic HCT can lead to longterm disease control in patients with high-risk MM

B. “ Long-term survival of 1338 MM patients treated with tandem autologous vs. autologous-allogeneic transplantation”, Costa et al, BMT 2020. This showed improved PFS and OS, and in post relapse survival.

3. In your conclusion, you cannot say “ we found” because you did not do any clinical work or clinical investigation. need to change the wording.

Author Response

Reviewer2

    1: This is a review of treatment strategies and yet there are no tables listing these treatments in any of the sections listed. It is very important that tables of studies are generated to understand the sections discussed

Recommend the following:

A: Table of studies looking at T-cells, NK cells, T-regs and macrophages assessment on multiple Myeloma (MM) cells/patients compared to normal- listing the studies, patients #s, treatment use, what measurement was done, what the values are, conclusions etc.

Response: Thank you for pointing out the need to add a table showing the function of immune cells. We have added table 1 ( a summary of function of immune cells and anti-myeloma agents for immune system), as shown below.

  1. Table that shows studies of improved immunological environment with the Imids, proteosome inhibitor(PIs), moabs, and ASCT, compared to control. Layout as in A

Response: Thank you for pointing out the need to add a table showing the immunotherapy agents. We have added table 1 ( a summary of anti-myeloma agents for immune system), as shown below.

Lines 137–138.

A summary of function of immune cells for healthy individuals and MM patients is shown in Table 1.

Lines 198–199.

A summary of anti-myeloma agents for immune system is shown in Table 1.

Lines 302–319.

CTL

NK cell

Treg

Breg

Macrophage

TAM

Function in healthy individuals

Adoptive immune responses activated by IL-2 and anti-tumor effect by releasing IFN-gamma [19]

Innate immune response, and regulate the activities of T cells, macrophages, and dendritic cells [34]

Suppression for immune response via immune tolerance by modulating antigen presentation, the consumption of IL-2, and the expression of negative regulatory cell surface receptors [35,41,42].

Regulation of immune responses via stimulation of IL-10 [49].

Various activities depending on the body site.

Function in MM patients

Exhaustion in tumor-site via upregulation of inhibitory receptors [22,24,26]

NK cell activity is reduced in patients with MM [35]

Elevated Treg level, predictors poor prognosis [43-45]. Transformation from non-Tregs into Tregs by secretion of ICOS-L [46].

Induction of an immunosuppressive BM microenvironment [50].

Relation with ADPC [51]

Suppress T-cell activity [54]. Induction of resistance to chemotherapy [55].

IMiDs

Activate via INF-gamma and IL-2 from T-cell from CD4+T-cell and PD-1 blockade [73, 78]

Activate ADCC by INF-gamma and IL-2 from CD4+T-cell [73-75]

Inhibit via down regulation of FOXP3 [76]

Decrease TAM via conversion from TAM into M1 macrophage [79]

Anti-CD38 MoAb

Induce clonal increasing CTL [84]

Decrease CD38+ NK cell, but activate ADCC by CD38-/low NK cell [83,85,86]

Inhibit activity via suppression ADO [88-90]

Inhibit activity via suppression ADO [88-90]

Activate ADCP [105]

Anti-PD-1 MoAb

Activate via PD-1 blockade [31]

Activate via PD-1 blockade [31]

Belantamab Mafodotin

Activate ADCC [96]

Activate ADCP [96]

HD-MEL + ASCT

Decrease T-cell but activate via activated dendritic cell by releasing HMGB1 [98,99]

Decrease Treg by HD-MEL [99,100]

PIs

Activate ADCC via HLA class1 blockade [101,102]

Table 1. Summary of function of immune cells and anti-myeloma agents for immune system. IMiDs, immunomodulatory drugs; MoAb, monoclonal antibody; PD-1, programed death-1; HD-MEL, high dose-melphalan; ASCT, autologous stem cell transplantation; PIs, proteasome inhibitor; CTL, cytotoxic T-cell; NK cell, natural killer cell; Treg, regulatory T-cell; Breg, regulatory B cell; TAM, tumor associated macrophage; MM, multiple myeloma; IL-2, interleukine-2; IFN-gamma, interferon gamma; ICOS-L, inducible T-cell co-stimulator ligand; ADCC, antibody-dependent cellular cytotoxicity; ADCP, antibody-dependent cellular phagocytosis; HMGB-1, high-mobility group box-1; FOXP3, forkhead box P3; ADO, adenosine.

  1. Table of studies in which MRD assessment was done in Newly dx MM and relapse pts with use of Novel agents (Imids, PIs, Moabs) and list outcomes of MRD neg vs MRD pos.

Response: Thank you for suggesting the need for a summary of results of phase 3 clinical trials investigating MRD status. We have provided the table, as shown below.

Lines 329–344.

A summary of phase 3 clinical trials investigating MRD status was shown in Table 2.

Trial

Disease status

Cutoff of MRD negativity (method)

Treatment

MRD negative rate

Outcome (MRD- vs MRD+)

IFM2009 [108]

TE-NDMM

10-6 (NGS)

VRd followed by ASCT followed by VRd vs VRd alone followed by LEN-MT.

ASCT arm 30%,

VRd alone arm 20%.

Median PFS: NR vs 29mo.

EMN02/HO95 [109]

TE-NDMM

10-5 (MCF)

CVD followed by VMP vs ASCT followed by VRd as CONS vs OBS followed by LEN-MT.

Post CONS: 76%.

5yr PFS: 79% vs 48%.

RPIMeR [110]

TE-NDMM

10-5(MCF)

Tandem ASCT vs single ASCT vs ASCT + 4cycles VRd.

After 1year post first ASCT MRD,

tandem ASCT 92%, single ASCT 78%, ASCT+VRd 85%.

PFS 76% vs 44%, OS 96% vs 66%.

FORTE [111]

TE-NDMM

10-5 (NGS)

KRd followed by ASCT followed by KRd (A) vs KRd 12 (B) vs KCd followed by ASCT followed by KCd (C).

Arm A: 42%, Arm B: 58%,  Arm C: 54% using NGS (10-5)

Arm A: 42%, Arm B: 58%,  Arm C: 54% using NGS (10-6).

CASSIOPEIA [112]

TE-NDMM

10-5 (MCF)

10-6 (NGS)

D-VTd followed by ASCT followed by D-VTd vs VTd followed by ASCT followed by VTd.

Post IND, D-VTd 35%, VTd 23%; Post CONS, D-VTd 64% vs VTd 44% using MCF (10-5); D-VTd 39% vs VTd 23% using NGS (10-6).

CASSIOPEIA2 [113]

TE-NDMM

10-5 (MCF)

10-6 (NGS)

D-MT vs OBS.

D-MT 6.61%, OBS 55.2% using MCF (10-5); D-MT 49.5%, OBS 36.7% using NGS (10-6).

MAIA [7]

NTE-NDMM

10-5 (NGS)

DRd vs Rd.

DRd 24.1%, Rd 7.3%.

ALCYONE [114]

NTE-NDMM

10-5 (NGS)

D-VMP followed by D-MT vs VMP alone.

D-VMP 28%, VMP 7%..

POLLUX [115]

RRMM

10-5 (NGS)

DRd vs Rd.

DRd 32.5%, Rd 6.7%.

DRd arm, NR vs 27.5mo;     

Rd arm, 55.3mo vs 15.7mo.

CASTOR [115]

RRMM

10-5 (NGS)

DVd vs Vd

DVd 15.1%, Vd 1.6%.

DVd arm, NR vs 12.4mo; Vd arm, 37.6mo vs 6.8mo

CANDOR [116]

RRMM

10-5 (NGS)

DKd vs Kd

DKd 18%, Kd 4%.

 Table 2. Summary of phase 3 clinical trials investigating MRD status. MRD, minimal residual disease; TE-NDMM, transplantation eligible newly diagnosed multiple myeloma; NTE-NDMM, not transplantation eligible newly diagnosed multiple myeloma; RRMM, relapsed or refractory multiple myeloma; NGS, next generation sequencing; MCF, multi-color flowcytometry; DARA, daratumumab; LEN, lenalidomide; K, carfilzomib (CFZ); V, bortezomib (BOR); T, thalidomide (THAL); M, melphalan (MEL); C, cyclophosphamide (CPA); d, dexamethasone (DEX); ASCT, autologous stem cell transplantation; CVd, CPA, BOR plus DEX; VMP, BOR, MEL plus prednisone; VRd, BOR, LEN plus DEX; VTd, BOR, THAL plus DEX; D-VTd, DARA plus VTd; KRd, CFZ, LEN plus DEX; KCd, CFZ, CPA plus DEX; Rd, LEN plus DEX; DRd, DARA plus Rd; VMP, BOR, MEL plus prednisone; D-VMP; DARA plus VMP; Vd, BOR plus DEX; DVd, DARA plus Vd; Kd, CFZ plus DEX; DKd, DARA plus Kd; CONS, consolidation therapy; MT, maintenance therapy; OBS, observation; PFS, progression free survival; OS, overall survival; NR, not reached; mo, months.

  1. in section 7: “Characteristics of residual MM cells in MRD positive patients”

The statement “--´However, some patients do not 329 achieve MRD negativity after intensive treatment due to residual myeloma cells that are 330 resistant to the administered agents. Such patients require a change in treatment strategy” cannot be true.

Can you please site studies in which patients with MRD+ after intensive treatment have had a change in their treatment strategies.  As of now, MRD status is not being used to change treatment strategies. For example, a patient in VGPR or CR but MRD+ and on Imid or PI maintenance, are there any studies looking at either changing their maintenance to another regimen vs leaving on len?  Please discuss these studies. As far as I know, there is no published randomized studies on this.  

The same applies for relapse patients who are on a certain regimen. Are their any studies in the relapse setting where MRD status  was used to change treatment vs continue same therapy?  If so please state.

It will also be beneficial for the authors to list ongoing studies looking at assessment of MRD status in newly Diagnosed patients  to determine if stopping maintenance in MRD- pts versus continuation of maintenance made a difference in PFS/OS- for eg the SWOG 1803

Response: Thank you for highlighting our misleading expression about MRD-adapted treatment. We completely agree with your opinion. “However, some patients do not achieve MRD negativity after intensive treatment due to residual myeloma cells that are resistant to the administered agents. Such patients require a change in treatment strategy.” This expression was exaggerated and, we have revised it as shown below.

Lines 407–422.

Some clinical trials concerning MRD status-adapted treatment strategies are currently ongoing (Table 3). The MASTER trial investigated the efficacy of four cycles of D-KRd (DARA, CFZ, LEN plus DEX) followed by ASCT and eight cycles of D-KRd consolidation treatment in achieving MRD negativity for NDMM [132]. If MRD negativity was achieved, chemotherapy was discontinued. Approximately 62–78% of patients achieved MRD negativity after the treatment, including eight cycles of consolidation therapy, and treatment could be discontinued. However, some patients do not achieve MRD negativity due to the presence of residual myeloma cells that are resistant to the administered treatment. Such patients might require a change of treatment strategy; however, to our best knowledge, no MRD status-adapted treatment strategy has been established. In addition, the clinical significance of pre-emptive therapy for RRMM with conversion from MRD negativity to positivity has not been analyzed to date. The PREDETOR trial is investigating the efficacy of pre-emptive therapy for patients with RRMM who have achieved MRD negativity after the last chemotherapy and may reveal the improvement of survival time of patients with RRMM whose MRD status converted to positivity [NCT03697655].

Lines 455-462.

Phase

Disease status

Study design

Primary endpoint

DRAMMATIC

[NCT04071457]

3

NDMM

DR versus LEN alone maintenance after ASCT (randomization1). If MRD+, maintenance Tx continues. If MRD-, maintenance Tx continue versus stop (randomization 2).

Overall survival between DR and LEN alone.

REMMANT

[NCT04513639]

3

RRMM

Salvage Tx, DKd until PD for the patients with loss MRD- (Arm A) versus PD according to IMWG criteria.

Progression free survival

MASTER

[NCT03224507]

2

NDMM

D-KRd 4 cycles followed by ASCT; consolidation Tx, D-KRd 8 cycles; maintenance Tx, LEN alone until PD. If MRD- achieved after ASCT, 4 or 8 cycles of D-KRd as consolidation Tx, treatment free observation.  

MRD- rate at the completion of consolidation Tx

DART4MM

[NCT03992170]

2

NDMM

DARA monotherapy every week in 1-8 weeks and every 2 weeks in 9-24 weeks. If MRD+, DARA every 4 weeks for 80 weeks; if MRD-, DARA stop.

Overall response rate

[NCT04140162]

2

NDMM

Induction Tx, DRd 1-24 weeks; consolidation Tx, D-VRD for only MRD+ 25-36weeks; maintenance Tx, DR 37-88 weeks followed by LEN alone until PD.

MRD- rate after induction and consolidation Tx

PREDATOR

[NCT03697655]

2

RRMM

Pre-emptive DARA until PD versus observation for MRD+ RRMM after MRD- by last line chemotherapy.

Event free survival

Table 3. Clinical trials concerning MRD status adapted treatment strategies. NDMM, newly diagnosed multiple myeloma; RRMM, relapsed or refractory multiple myeloma; MRD, minimal residual disease; PD, progressive disease; Tx, treatment; DARA, daratumumab; LEN, lenalidomide; K, carfilzomib (CFZ); V, bortezomib (BOR); d, dexamethasone (DEX); ASCT, autologous stem cell transplantation; DR, DARA plus LEN; DRd, DARA, LEN plus DEX; DKd, DARA, CFZ plus DEX; D-KRd, DARA, CFZ, LEN plus DEX; D-VRd, DARA, BOR, LEN plus DEX; IMWG, international myeloma working group.

  1. Recommend an algorithm figure for MM treatment to improve immunological environment and MRD negativity in newly dx MM pts and in relapse patients

 Response: Thank you for the recommendation to add an algorithm for MM treatment. We have added the algorithm of the treatment strategy for NDMM in “section 11 Future directions,” as shown below. On the other hand, we could not add a specific algorithm of the treatment strategy for RRMM because this was very complicated considering prior treatments. Therefore, we described a concept for salvage chemotherapy concerning MRD status and immune environment in figure 3.

Lines 628–644.

Treatment algorithm concerning MRD and immunological status for NDMM when anti-CD38 MoAb, IMiDs, and PIs are available was shown in figure 2.

Figure 2. Treatment algorithm concerning MRD and immunological status. Combination therapy with anti-CD38 MoAb, IMiDs, PIs, and ASCT may be suitable for MM patients considering the efficacy against myeloma cells and improved immune system. If the MRD status is negative, the current treatment should continue. However, if the MRD status is positive, the genetic mutation of residual myeloma cells should be analyzed to optimize treatment. If the mutation related to resistance for IMiDs or PIs is detected, treatment should be changed. If the mutation related the resistance for IMiDs or PIs is not detected, agents for activation of the immune system might be added. MRD assessment should be repeated in the patients with MRD negativity. If MRD status converts to positivity in PB sample, imaging techniques should be evaluated to detect EMD. Thereafter, the current Tx could continue with close monitoring of MRD status. MRD, minimal residual disease; IMiDs, immunomodulatory drugs; PIs, proteasome inhibitors; MoAbs, monoclonal antibodies; ASCT, autologous stem cell transplantation; Tx, treatment; PB, peripheral blood; EMD, extramedullary disease.

Minor:

  1. please make sure font and font size is uniform through out.

Response: Thank you for your comment. We have ensured the font and font size are uniform throughout the manuscript.

  1. in section 9 “ Immunological treatment to eradicate residual MM cells”, the authors forget to site and discuss these studies:

A: “Tandem Autologous-Autologous versus Autologous-Allogeneic Hematopoietic Stem Cell Transplant for Patients with Multiple Myeloma: Long-Term Follow-Up Results from the Blood and Marrow Transplant Clinical Trials Network 0102 Trial”,  by Giralt et al-  BBMT, 2019, 798-804.  That showed Allogeneic HCT can lead to longterm disease control in patients with high-risk MM

  1. “ Long-term survival of 1338 MM patients treated with tandem autologous vs. autologous-allogeneic transplantation”, Costa et al, BMT 2020. This showed improved PFS and OS, and in post relapse survival.

 Response: Thank you for pointing out additional literature about immunological treatment. We have added two previous articles on the benefit of allogeneic stem cell transplantation according to reviewer’s recommendation, as shown below.

Lines 512–514.

Two clinical trials have revealed long survival of patients treatment with ASCT followed by allogeneic HSCT compared to those treated with tandem ASCT [150, 151].

  1. In your conclusion, you cannot say “ we found” because you did not do any clinical work or clinical investigation. need to change the wording.

Response: Thank you for pointing out the mistake about our wording. We have changed the expression, as shown below.

Lines 686–687.

We found that improvement of the immune environment and maintenance of MRD negativity are key factors for the long-term survival of MM patients.

->

We consider that improvement of the immune environment and maintenance of MRD negativity are key factors for the long-term survival of MM patients.

Reviewer 3 Report

The Authors provide a well written and complete review on therapeutic strategy for improving the immunological environment and eradicating minimal residual disease (MRD).

I recommend the publication in present form

Author Response

Reviewer3

The Authors provide a well written and complete review on therapeutic strategy for improving the immunological environment and eradicating minimal residual disease (MRD).

I recommend the publication in present form

Response: Thank you for reviewing our manuscript and for the positive feedback.
